# Effect of the Heterovalent Doping of TiO$_2$ with Sc$^{3+}$ and Nb$^{5+}$ on the Defect Distribution and Photocatalytic Activity

Petr D. Murzin, Aida V. Rudakova , Alexei V. Emeline and Detlef W. Bahnemann *

Laboratory "Photoactive Nanocomposite Materials", Saint-Petersburg State University,
199034 Saint Petersburg, Russia; murzinpetrff@gmail.com (P.D.M.); aida.rudakova@spbu.ru (A.V.R.);
alexei.emeline@spbu.ru (A.V.E.)
* Correspondence: detlef.bahnemann@spbu.ru

**Abstract:** Two series of Sc$^{3+}$- and Nb$^{5+}$-doped TiO$_2$ (rutile) samples were synthesized and characterized by SEM, ICPE spectroscopy, XPS, and BET methods. Photocatalytic activity of the doped TiO$_2$ samples was tested in photocatalytic degradation of phenol. Dependences of the photocatalytic activities of the doped TiO$_2$ samples demonstrate a volcano-like behavior, indicating the existence of the optimal dopant concentrations to achieve the highest activity of photocatalysts. Remarkably, the optimal dopant concentrations correspond to the extrema observed in work function dependences on the dopant concentrations, that indicates a significant energy redistribution of the defect states within the bandgap of TiO$_2$. Such a redistribution of the defect states is also proven by the alterations of the optical and EPR spectra of the intrinsic Ti$^{3+}$ defect states in TiO$_2$. Based on the analysis of the experimental results, we conclude that both Sc$^{3+}$ and Nb$^{5+}$ doping of TiO$_2$ results in redistribution of the defect states and the optimal dopant concentrations correspond to the defect structures, which are ineffective in charge carrier recombination, that ultimately leads to the higher photocatalytic activity of doped TiO$_2$.

**Keywords:** heterogeneous photocatalysis; photocatalytic activity; phenol degradation; metal doping; heterovalent doping; work function; intrinsic defects; photoinduced defect formation; titanium dioxide

## 1. Introduction

For the last decades, the metal doping of photocatalysts has become a standard approach to modify the electronic and optical properties of photoactive materials, and therefore, to change their photocatalytic behavior [1–10]. However, in spite of the quite long history of the doping effect exploration in heterogeneous photocatalysis, only two systematic approaches to analyze and describe the effects of various dopants on photocatalytic activity of photoactive materials have been developed until now. The first approach is based on the electronic theory of catalysis developed in the 1960s [11–13]. According to the theory, the activity of catalysts in chemical reactions depends on the position of the Fermi level in semiconductor materials (see Figure 1a).

It assumes that the interaction of the adsorbed molecules with the semiconductor surface results in formation of the corresponding localized energy level of adsorbate complex within the bandgap of the semiconductor catalyst. Consequently, if the Fermi level position is located higher than the adsorbate complex level, the electron transfer from the semiconductor to adsorbed molecules occurs. In contrast, should the Fermi level be located below the adsorbate level, the electron transfer is directed from the adsorbed molecules to the semiconductor. The different directions of the electron transfer dictated by the Fermi level position result in the formation of different forms of the adsorbed species, and therefore in their different chemical reactivity. A typical pathway to change the Fermi level position in the semiconductor is heterovalent doping of the semiconductor. Indeed, semiconductor doping with electron-donor atoms results in a shift of the Fermi level toward

the conduction band, whereas substitution of the lattice host atoms with electron-acceptor dopants leads to a Fermi level toward the valence band. Thus, heterovalent doping is considered as a method to change the Fermi level position, and therefore to manipulate the chemical reactivity of the adsorbed molecules.

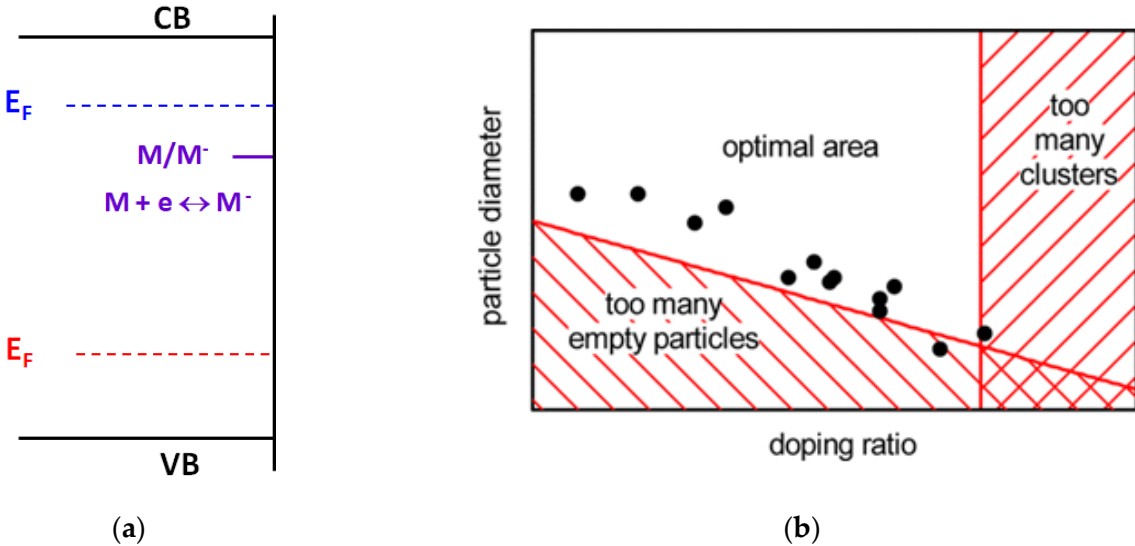

(a)  (b)

**Figure 1.** (**a**) Illustration of basic postulates of the electronic theory of catalysis. (**b**) Illustration of the statistical approach for the optimal doping concentration, adapted with permission from [14]. Copyright 2012 American Chemical Society.

Another approach based on statistical analysis of the large array of experimental data of photocatalytic activity accumulated for various photocatalysts doped with different metals (see Figure 1b) was proposed by Bloh et al. [14] ten years ago. Based on the analysis of the particle size distribution and dopant concentration, it was concluded that a higher activity of photocatalysts is observed for those samples in which particles contain a single dopant atom per particle. Therefore, there is an optimal dopant concentration secured for this condition: at lower concentrations, there are "empty" particles without dopants, and at higher concentrations, dopant atoms (ions) form dopant clusters which can act as efficient recombination centers. Though it was not mentioned in the original article, the statistical analysis in the model was performed only for the photocatalysts containing heterovalent dopants. Note that both approaches are too general and do not consider both chemical and physical peculiarities of dopant atoms (ions), and are focused only on either thermodynamical or statistical effects.

In this study, we performed systematic research of the effect of heterovalent doping of $TiO_2$ (rutile) with $Sc^{3+}$ and $Nb^{5+}$ cations at different dopant concentrations to understand the reasons for the dopant-induced alteration of the photocatalytic activity and existence of the dopant optimal concentration for higher activity of the photocatalyst. The selection of the heterovalent dopant cations, $Sc^{3+}$ and $Nb^{5+}$, was dictated by the following reasons: (i) substitution of the host $Ti^{4+}$ cations with either $Sc^{3+}$ or $Nb^{5+}$ creates an excess of either negative or positive charge, respectively, in the lattice, which must be compensated due to formation of the corresponding intrinsic defects, and (ii) the energy levels of 3d and 4d electronic states of $Sc^{3+}$ and $Nb^{5+}$, respectively, are higher than 3d levels of $Ti^{4+}$, and therefore are located in the conduction band and do not create energy levels within the bandgap of the host $TiO_2$. Thus, one can observe the redistribution of only the intrinsic defects within the bandgap induced by heterovalent doping with either $Sc^{3+}$ or $Nb^{5+}$. At the same time, it is wise to note that lattice distortion responsible for the formation of the intrinsic defect states can also be induced by doping due to differences in the ion radii, which is slightly larger for $Nb^{5+}$ (0.680 Å) and significantly larger for $Sc^{3+}$ (0.745 Å) dopants compared to the radii of the host $Ti^{4+}$ (0.605 Å) cations.

The optical properties of Nb-doped titanium dioxide do not differ much from those of pristine $TiO_2$, which determines its application in the field of transparent conductive oxides [15]. Most studies either do not detect alteration of the spectral position of the absorption edge of Nb-doped $TiO_2$ or observe a blue shift due to the Burstein–Moss effect [16–18]. The position of intrinsic defect states associated with niobium dopant within the bandgap of titanium dioxide depends on $TiO_2$ phase. The formation of shallow states at the edge of the conduction band with electron density distributed over several titanium atoms was demonstrated in anatase phase by DFT calculations [19]. At the same time, when niobium was introduced into rutile, defect states were located close to the middle of the bandgap and electron density was significantly localized and distributed only between two neighboring titanium atoms. These deep states were experimentally observed by deep-level transient spectroscopy [20] and X-ray photoelectron spectroscopy [21].

Photocatalytic activity of Nb-doped titanium dioxide has been explored for various reactions. For instance, an increase of $CO_2$ conversion to methanol was observed for Nb-doped (0 to 0.65 at.%) anatase [22]. A series of Nb-$TiO_2$ (anatase) samples with the dopant concentration varied up to 10 at.% were synthesized and explored in methylene orange bleaching reaction, and the sample with 0.5 at.% Nb content demonstrated the highest photoactivity [23]. The same sample also showed the highest hydrogen generation rate after Pd deposition. It was demonstrated in [24] that UV irradiation results in a faster transition of Nb-doped $TiO_2$ films to a super-hydrophilic state compared with undoped $TiO_2$. Typically, the selected concentration limits hinder identification of the optimal dopant concentration, which leads to incorrect conclusions about the effect of niobium on the photoactivity of $TiO_2$. For example, in [25], Nb-doped $TiO_2$ demonstrated a continuous increase of photoactivity towards methylene blue decomposition in a concentration range from 0 to 0.59 at.%. At the same time, in another study [16], it was shown that the decomposition rate of methylene blue decreased within the concentration range of 0.74–2.67 at.%. Therefore, such series of Nb-doped $TiO_2$ samples do not allow to establish an optimal dopant concentration.

Similar to niobium doping, scandium doping changes the optical properties of $TiO_2$ insignificantly. There are experimental [26,27] and theoretical [28] studies that show some broadening of the optical bandgap for Sc-$TiO_2$. However, in all cases, researchers report rather small changes in the position of the absorption edge.

Just a few papers report studies on the photocatalytic activity of scandium-doped titanium dioxide. For instance, in [29], Sc-doped $TiO_2$ films were synthesized in the anatase phase at concentrations of 0.5%, 2%, and 5%, and a slight change of the absorption edge was noted. The $TiO_2$ sample with a scandium concentration of 5 at.% demonstrated an increased rate of diclofenac potassium decomposition, while at 0.5 at.%, a decrease of photoactivity was observed. In an extensive study of the effect of 35 dopants on the behavior of Grätzel cells [30], it was noted that 2 at.% of scandium dopant significantly attenuates the light conversion efficiency.

Thus, to understand the major reasons for the dopant-induced alteration of the photocatalytic activity and to establish if there exists an optimal dopant concentration to achieve higher activity of the $TiO_2$ photocatalysts, more systematic and detailed studies with a wide range of dopant concentrations are required.

## 2. Results and Discussion

Photocatalytic phenol degradation was chosen as a test reaction and the initial rate of phenol degradation was taken as a parameter characterizing the photoactivity of the studied $TiO_2$ photocatalyst samples. Figure 2 demonstrates the dependences of the $TiO_2$ photocatalysts' activities on the dopant concentrations: either $Sc^{3+}$ (Figure 2a) or $Nb^{5+}$ (Figure 2b).

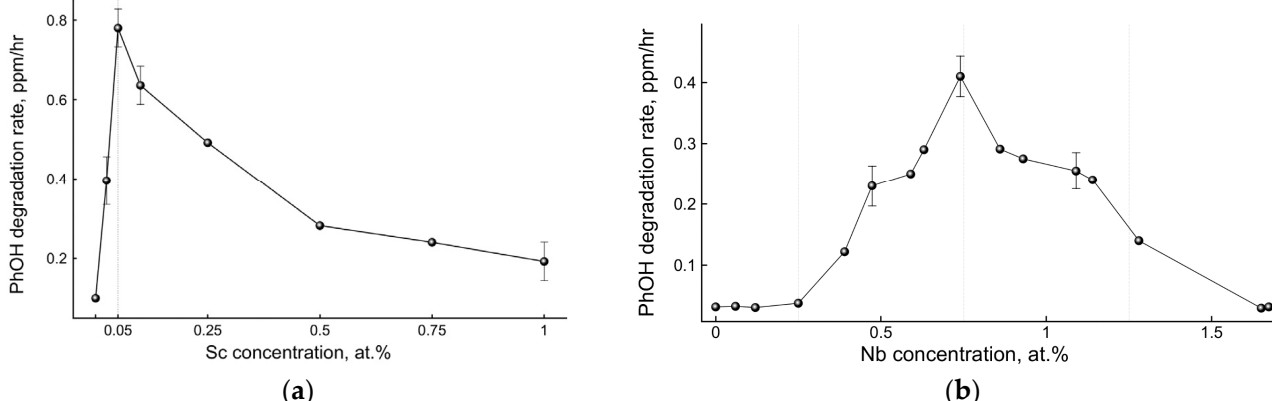

**Figure 2.** Experimental dependences of the initial rates of phenol photocatalytic degradation on dopant concentrations in $TiO_2$: (**a**) $Sc^{3+}$ and (**b**) $Nb^{5+}$.

Both dependences clearly demonstrated a volcano-like behavior with well-pronounced maximal extrema at certain values of dopant concentrations. It is wise to note that the $Sc^{3+}$ and $Nb^{5+}$ dopant concentrations corresponding to the maximal activity of doped $TiO_2$ were different, which indicates the importance of the dopant nature. Moreover, the dopant concentrations corresponding to the maximal photocatalytic activity were significantly larger ($10^5$–$10^7$ dopant ions per particle) than those predicted by the statistical approach mentioned above (see Figure 1b) [14]. Therefore, the effect of the dopants on the activity of photocatalysts is more complex than a dopant number factor.

Figure 3 demonstrates the alteration of the work function of doped $TiO_2$ samples depending on the $Sc^{3+}$ and $Nb^{5+}$ dopant concentrations, together with the dependences of photocatalytic activities shown in Figure 2.

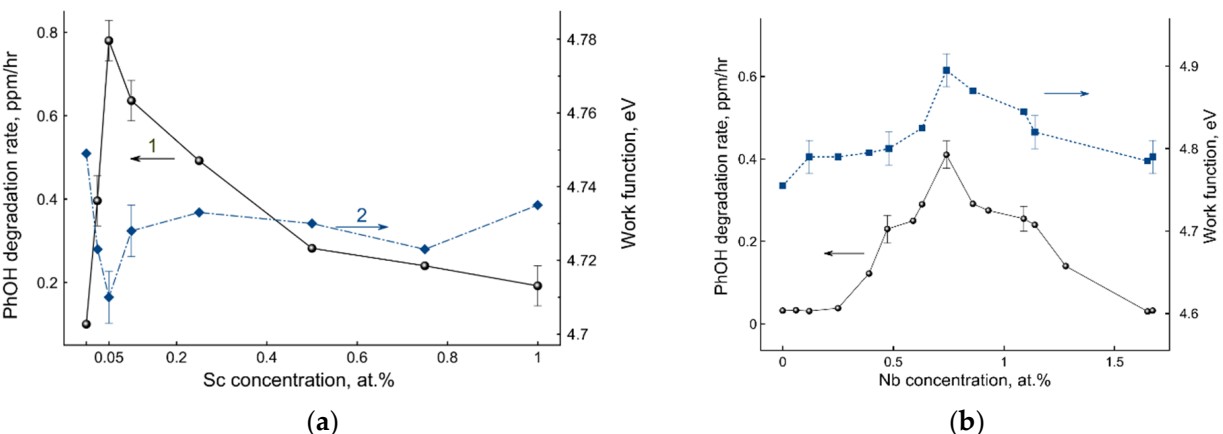

**Figure 3.** Dependences of the photocatalytic activities (curves 1, in black) and work functions (curve 2, in blue) of doped $TiO_2$ on dopant concentrations: (**a**) $Sc^{3+}$ and (**b**) $Nb^{5+}$.

The dependences of work functions on $Sc^{3+}$ and $Nb^{5+}$ dopant concentrations clearly demonstrated the apparent correlations with dependences of photocatalytic activities, and their extrema were observed at the same values of dopant concentrations. At the same time, the characteristics of work function behavior observed for $Sc^{3+}$ and $Nb^{5+}$ dopants were different: for $Sc^{3+}$ doping, the maximum of photocatalytic activity corresponds to the minimum of the work function, while for $Nb^{5+}$ doping, the maximum of photocatalytic activity correlates with the maximum of the work function value. This observation contradicts the prediction of the electronic theory of catalysis due to the different trends in the alteration of work functions observed for $Sc^{3+}$ and $Nb^{5+}$ dopants, and therefore, different

trends in the alteration of the Fermi level position result in a maximum of photocatalytic activity in both cases.

In general, the alteration of the work function, indicating the alteration of the Fermi level, originates from either the appearance of the new electronic states or the rearrangement or redistribution of the existing electronic states within the bandgap of the semiconductor. A Fermi level shift toward the conduction band originates from the higher concentration of the electron-donor states located close to the bottom of the conduction band, while a Fermi level shift toward the valence band means higher concentrations of the electron-acceptor states at the top of the valence band. Thus, one may conclude that $TiO_2$ doping with either $Sc^{3+}$ or $Nb^{5+}$ cations at optimal concentrations results in redistribution of the various defect states, leading to a Fermi level shift toward either the conduction band or the valence band, respectively, that in turn corresponds to the maximal photocatalytic activity for both types of doped photocatalysts.

To confirm such dopant-induced defect state redistributions and to establish the corresponding types of defect states, we performed spectroscopic studies of photoinduced defect formation by means of both diffuse reflectance spectroscopy and EPR spectrometry. Figure 4 demonstrates the absorption spectra of photoinduced defects generated during irradiation for 20 min for the samples with different dopant concentrations.

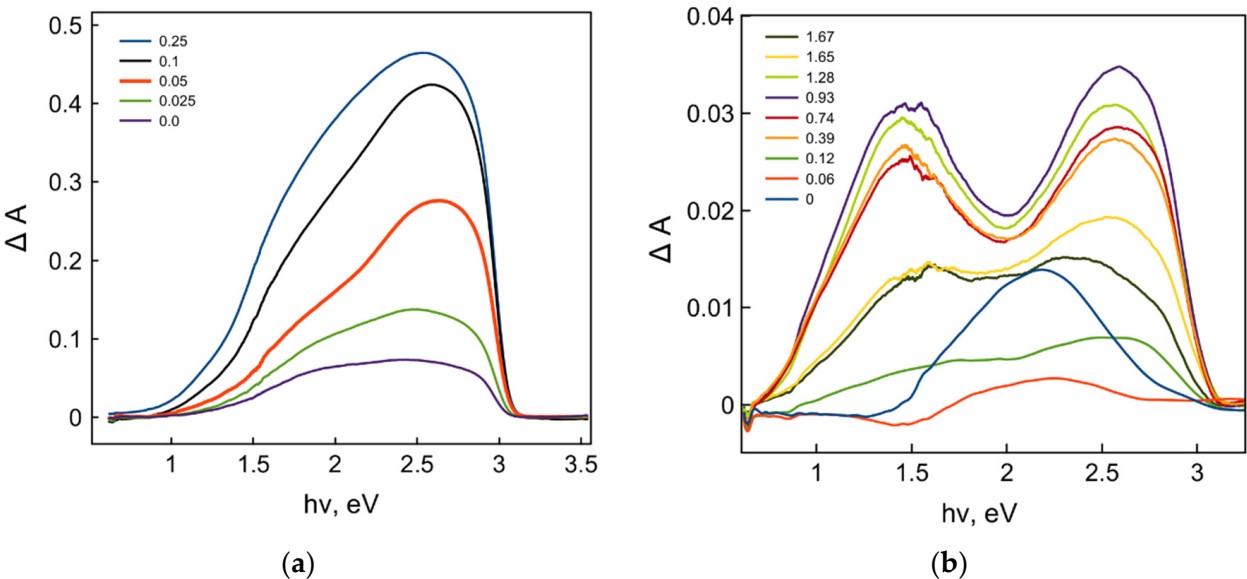

(a)                                                                              (b)

**Figure 4.** Absorption spectra of photoinduced defects generated for 20 min for the $TiO_2$ samples with different dopant concentrations: (**a**) $Sc^{3+}$ and (**b**) $Nb^{5+}$.

A detailed description of the mechanism of photoinduced defect formation can be found elsewhere [31–33] and its simplified version is presented in the Supplementary Materials (see also Figure S1).

A simple visual comparison of two sets of photoinduced defect absorption spectra for $Sc^{3+}$- and $Nb^{5+}$-doped $TiO_2$ samples inferred that the absorption values for both sets of the samples strongly depend on the dopant concentrations. Moreover, the shapes of the defect absorption spectra were rather different, which indicates the different impact of the different defect states on the absorption spectra, that is a manifestation of the defect redistribution. Figure 5 demonstrates the deconvolutions of the photoinduced defect absorption spectra with the three single absorption bands resulting in the best fitting of the original spectra.

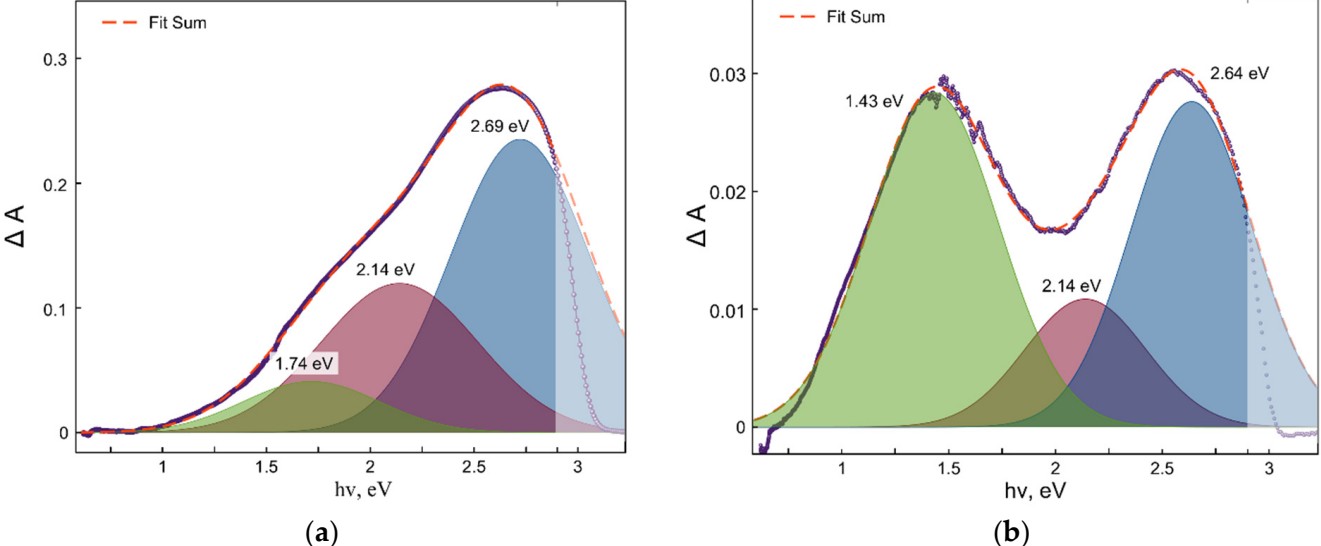

**Figure 5.** Deconvolution (red dashed lines) of the absorption spectra (blue solid lines) of photoinduced defects in (**a**) Sc-doped $TiO_2$ and (**b**) Nb-doped $TiO_2$.

Remarkably, the deconvolution of the absorption spectra of photoinduced defects for both Sc- and Nb-doped $TiO_2$ resulted in the best fitting with three single Gaussian bands, the parameters of which (energy of maxima) correspond well to the absorption bands of the different $Ti^{3+}$ intrinsic defect states observed earlier elsewhere [34,35]. Thus, the results of the deconvolution of both absorption spectra of photoinduced defects clearly indicated that both $Sc^{3+}$ and $Nb^{5+}$ doping of $TiO_2$ caused significant redistribution of the intrinsic defect states in titanium dioxide and did not create new types of defects.

Both pre-existing and photoinduced $Ti^{3+}$ defect states can also be detected by the EPR spectrometry method. Supplementary Figure S2 demonstrates EPR spectra of the intrinsic paramagnetic defects in pristine and in $Sc^{3+}$- and $Nb^{5+}$-doped $TiO_2$.

EPR spectra of Sc-doped $TiO_2$ consist of groups of signals with the most intense bands at g = 1.905, 1.940, and 1.965 (Supplementary Figure S2b). According to previously published results, the signals should be attributed to $Ti^{3+}$ centers in rutile [36,37]. Depending on the dopant concentration, EPR spectra of Sc-doped $TiO_2$ demonstrated both qualitative and quantitative redistribution of the defect states. Most clearly, this redistribution of $Ti^{3+}$ defect states can be seen in EPR absorption spectra obtained by integration of the original EPR spectra (Supplementary Figure S3).

EPR spectra of Nb-doped $TiO_2$ samples are shown in Supplementary Figure S2c. The spectra are presented by a complex signal in the region of g~1.94, which can be attributed to $Ti^{3+}$ defect states in rutile [36,37]. Nb doping leads to higher amounts of $Ti^{3+}$ states with respect to the undoped sample, that can be attributed to the electron-donor behavior of niobium states in titanium dioxide lattice.

The effect of UV irradiation on EPR spectra of pristine and $Sc^{3+}$- and $Nb^{5+}$-doped $TiO_2$ (Supplementary Figure S2) indicates that photoexcitation of $TiO_2$ results in a further redistribution of the defect states due to charge trapping and recombination, which correlates qualitatively with the results obtained by optical spectroscopy (Figure 4).

The double integral of the EPR signal is proportional to the number of detected paramagnetic defect states. Figure 6 demonstrates the dependences of EPR double-integral values on the $Sc^{3+}$ and $Nb^{5+}$ dopant concentrations after UV irradiation for 20 min.

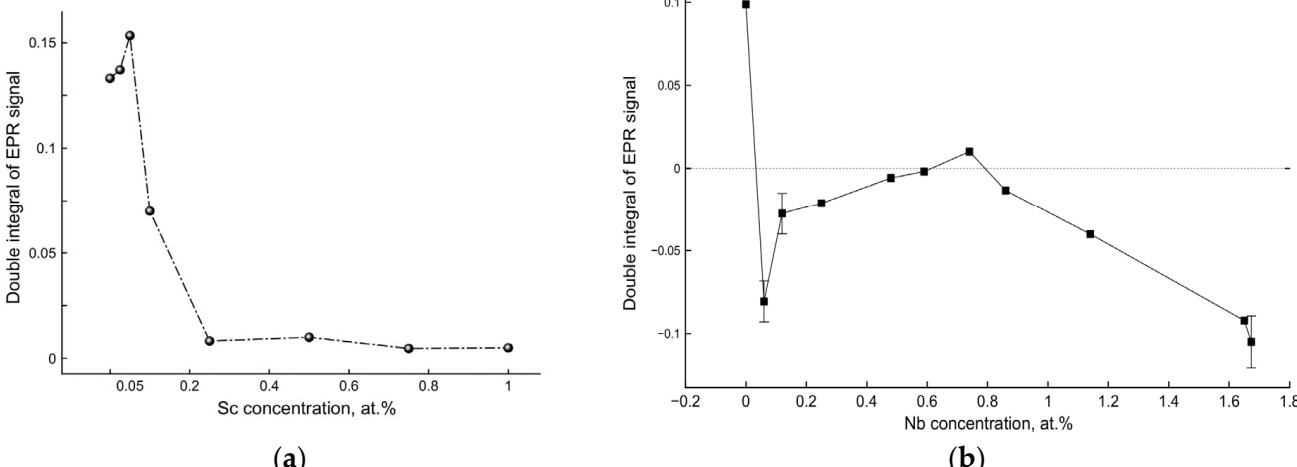

**Figure 6.** Dependences of the double-integrated EPR signal corresponding to the number of photoinduced paramagnetic $Ti^{3+}$ states on the dopant concentrations in: (**a**) Sc-doped $TiO_2$ and (**b**) Nb-doped $TiO_2$.

Dependences of the double-integrated EPR signal on the $Sc^{3+}$ and $Nb^{5+}$ dopant concentrations clearly demonstrate very different behavior with extrema corresponding to the dopant concentrations, resulting in the highest photocatalytic activities of the doped $TiO_2$ samples. Indeed, the EPR signal of Sc-doped $TiO_2$ indicates that UV irradiation results in an increase of the number of $Ti^{3+}$ states (most pronounced for the optimal Sc dopant concentration in terms of photocatalytic activity), whereas for Nb-doped $TiO_2$, UV irradiation results in a significant decay of the number of $Ti^{3+}$ states compared to undoped $TiO_2$, and the number of these states remains stable only at the optimal Nb dopant concentration corresponding to the sample with the highest photocatalytic activity. In other words, UV irradiation of Sc-doped $TiO_2$ results in the formation of new $Ti^{3+}$ states due to electron trapping by pre-existing defects, and UV irradiation of Nb-doped $TiO_2$ leads to the decay of $Ti^{3+}$ states caused by recombination with photogenerated holes, indicating in both cases the defect states' redistribution depending on both $Sc^{3+}$ and $Nb^{5+}$ dopant concentrations.

Considering the defect redistribution in $TiO_2$ depending on the dopant type and concentration, as indicated by diffuse reflectance spectra and EPR spectra, and indicating that the maximal or minimal effect of defect state redistribution is observed at the same dopant concentrations, which corresponds to the extrema in work function and photocatalytic activity, we propose the following mechanisms of the dopant effects on the photocatalytic activity of doped $TiO_2$.

$Sc^{3+}$ doping brings an excess of the negative effective charge into titanium lattice, and therefore creates a local electric field of the negative charge, which increases the energy of the electronic states of the intrinsic defect, making them shallow. This charge excess shall be compensated by the intrinsic defects possessing an effective positive charge, such as anion vacancies [38]. Since for low Sc dopant concentrations up to the optimal (0.05 at.%) the EPR signal of $Ti^{3+}$ centers increases (Figure 6), it is wise to assume the formation of defect states such as $Sc^{3+}-Vac_O-Ti^{3+}$ (Figure 7) with a compensated charge. At higher $Sc^{3+}$ concentrations, the formation of defect states such as $Sc^{3+}-Vac_O-Sc^{3+}$ is more favorable due to the larger number of dopant atoms, which is confirmed by a decrease of the $Ti^{3+}$ EPR signal and by a drastic alteration of absorption of the photoinduced defects (see Figures 4 and 6).

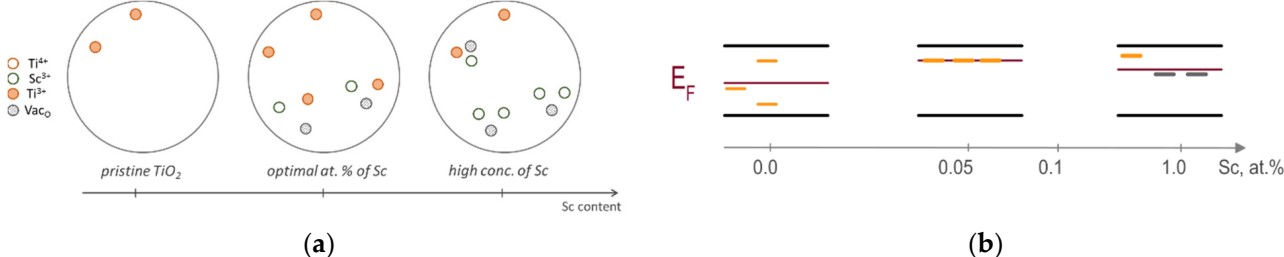

**Figure 7.** Illustration of the (**a**) spatial and (**b**) energy redistribution of the defect states in Sc-doped TiO$_2$. Yellow dashes are Ti$^{3+}$ centers, gray are deep centers of various types.

The spatial and structural redistribution of the intrinsic defects induced by Sc$^{3+}$ doping of TiO$_2$ is accompanied by corresponding energy redistribution of these defects within the bandgap of titanium dioxide. For Sc dopant concentrations below 0.05 at.%, EPR and optical spectra indicate an increase of the number of Ti$^{3+}$ states. Lower work function values imply the proximity of the position of the energy levels for these states to the conduction band. Such shallow traps are typically ineffective as recombination centers [39]. Stabilization of the shallow traps along with destabilization of intrinsic deep states (due to the presence of the negatively charged Sc$^{3+}$ states) lead to a decrease of the recombination efficiency, and therefore result in the promotion of photocatalytic activity of TiO$_2$. A further increase of the scandium concentration leads to a qualitative alteration in the structure of defect states, which is manifested by a significant decrease of the EPR signal of Ti$^{3+}$ centers, apparently due to the formation of Sc$^{3+}$–Vac$_O$–Sc$^{3+}$ defect states that stabilize deep states such as anion vacancies. In addition, an increase in the number of dopant atoms suggests the formation of deep defect agglomerates, which as a rule are effective in recombination processes. Thus, the formation of deep defect states at higher Sc dopant concentrations leads to a downward shift of the Fermi level position, an increase of recombination efficiency, and therefore, to the decrease of photocatalytic activity.

In turn, TiO$_2$ doping with Nb$^{5+}$ brings an excess of the positive charge into the lattice, and therefore creates the local electric fields of the positive charge, which decrease the energy of the electronic states of the intrinsic defects, making them deeper. At low concentrations, the excess of the positive charge of the Nb$^{5+}$ dopant states can be compensated by intrinsic Ti$^{3+}$ defects. For those Nb$^{5+}$ atoms that are located close to compensating Ti$^{3+}$ states, it is energetically favorable to form a defective structure such as Nb$^{5+}$–O$^{2-}$–Ti$^{3+}$ [40] (see Figure 8a). In this case, an electron transfer from the Ti$^{3+}$ cation to the Nb$^{5+}$ cation may also result in the formation of Nb$^{4+}$–O$^{2-}$–Ti$^{4+}$ complex defect states, which leads to a decrease of the corresponding EPR signal (see Figure 6b). Note that both types of defect states are electrically neutral and stable. At the optimal niobium dopant concentration (0.74 at.%), a number of such defect states approach the maximum extremum. However, a further increase of niobium content leads to an increased probability that Nb dopant atoms are located near the Nb$^{5+}$–O–Ti$^{3+}$ defect states, and thereby, their charge neutrality will be destabilized.

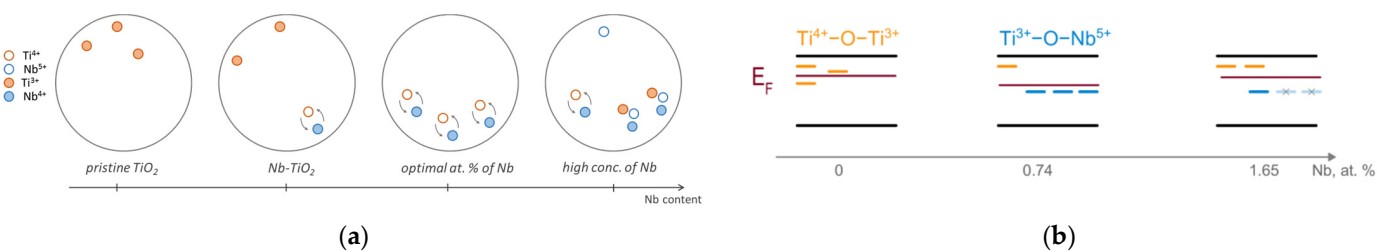

**Figure 8.** Illustration of the (**a**) spatial and (**b**) energy redistribution of the defect states in Nb-doped TiO$_2$. Yellow dashes are Ti$^{4+}$–O–Ti$^{3+}$ states, blue dashes are Ti$^{3+}$–O–Nb$^{5+}$ states.

The dependence of the work function on the niobium concentration indicates that the deep defect states prevail in the Nb-doped $TiO_2$ sample with the maximal photocatalytic activity (see Figure 8b). Due to their electroneutrality, such deep defect states as $Nb^{5+}$–O–$Ti^{3+}$ or $Nb^{4+}$–$O^{2-}$–$Ti^{4+}$ cannot behave as efficient centers of recombination in spite of the deep position of their energy levels within the bandgap. Thus, defect redistribution in the region of optimal concentrations leads to prevalence of such defects and results in a decrease of the charge carrier recombination rate, and hence to the promotion of the photocatalytic activity of $TiO_2$.

Summing up the experimental results, we conclude that both $Sc^{3+}$ and $Nb^{5+}$ doping of $TiO_2$ results in redistribution of the defect states, as proven by work function, optical, and EPR measurements, and the optimal dopant concentrations correspond to the defect structures, which are ineffective in charge carrier recombination, that ultimately leads to the higher photocatalytic activity of doped $TiO_2$.

## 3. Materials and Methods

### 3.1. Synthesis and Characterization

Two sets of $Sc^{3+}$- and $Nb^{5+}$-doped $TiO_2$ samples were synthesized by the solid-state thermochemical method. Original metal oxide powders ($TiO_2$ (anatase), $Nb_2O_5$, $ScCl_3$) with purity not less than 99.98% (Sigma-Aldrich, Saint Louis, USA & Vekton, Saint Petersburg, Russia) were used in a stoichiometric proportion, mixed, and ground with agate mortar in isopropyl alcohol for 40 min. The obtained mixtures were subsequently heated at 650, 800, and 1000 °C in the ambient atmosphere for 10, 10, and 25 h at each temperature, respectively, with intermediate milling after every heating cycle. After heating at 1000 °C, the samples were cooled down to room temperature with a temperature decay rate of 1°/min. The treatment time at 1000 °C was chosen to achieve a complete transition of titanium dioxide to the rutile phase. As a result, sixteen Nb-doped $TiO_2$ samples with the dopant concentration (x) ranging from 0.0 to 1.7 at.% (x (at.%) = 0.00, 0.06, 0.12, 0.25, 0.39, 0.48, 0.59, 0.63, 0.74, 0.86, 0.93, 1.09, 1.14, 1.28, 1.65, 1.67) and eight samples of Sc-doped $TiO_2$ with the dopant concentration ranging from 0.0 to 1.0 at.% (x (at.%) = 0.00, 0.025, 0.05, 0.10, 0.25, 0.50, 0.75, 1.00) were obtained. The dopant concentrations in the samples were determined by ICP atomic emission spectroscopy with a Shimadzu ICPE-9000 spectrometer (Kyoto, Japan).

Phase composition was determined by X-ray diffraction analysis. The "Rigaku Miniflex II" diffractometer was used with a Cu Kα emission line (anode voltage of 10 mA, accelerating potential difference of 30 kV) in the range of $10° \leq 2\theta \leq 80°$ and a scanning rate of 0.5°/min. Phase composition was determined by the Rietveld method using the TOPAS 4.2 program (Bruker AXS, Brisbane, Australia) linked to the ICSD database. According to XRD analysis, all samples were in rutile crystal phase (see Supplementary Figure S4). Microstructural images of and element distribution in the synthesized samples were obtained by means of the Zeiss Merlin scanning electron microscope. According to the electron microscopic images (see Supplementary Figure S5), all dispersed samples consisted of well-crystalized particles of a parallelepiped-like shape with a mean particle size of about 1.5–2.0 μm. The specific surface area measured by the BET method with nitrogen was about $1.0 \pm 0.2$ $m^2$.

X-ray photoelectron spectra were recorded by a Thermo Fisher Scientific Escalab 250Xi spectrometer (Waltham, MA, USA) (see Supplementary Figure S6). In addition to the typical XPS signals corresponding to $TiO_2$ (O1s, $Ti2p_{3/2}$, and $Ti2p_{1/2}$), the dopant signals for $Sc^{3+}$ ($Sc2p_{3/2}$) and $Nb^{5+}$ ($Nb3d_{3/2}$ and $Nb3d_{5/2}$ states, respectively) were clearly observed and their intensity increased with the increase of the corresponding dopant concentrations.

### 3.2. Photocatalytic Measurements

Photocatalytic activity of either Sc- or Nb-doped $TiO_2$ samples was tested in a reaction of phenol photodegradation over $TiO_2$. The initial rate of phenol decomposition was selected as a measure of photocatalytic activity. The photocatalytic reactor was a

400 mL glass, silver-coated on the outside with a lateral quartz window for illumination of suspension. Stirring of the reaction mixture was carried out with a magnetic stirrer. Standard experimental conditions were as follows: the initial phenol concentration was 100 ppm, the photocatalyst loading was 2 g/L, and the mixture volume was 300 mL. The solution was acidified to pH 3 by HCl. The pH value was selected in order to obtain phenol completely in molecular form and to avoid uncertainties associated with the surface charge state of $TiO_2$ (isoelectric point), and to stay within the region of independence of the phenol decomposition rate on pH [41]. Before irradiation, the solution was stirred in the dark for 1 h to achieve adsorption equilibrium. Blank experiments without $TiO_2$ were performed, demonstrating the absence of any changes in the phenol concentration over time due to direct phenol photolysis.

An Osram 150 W xenon lamp was used as the light source. In order to avoid direct phenol photolysis, the BS-6 cut-off filter (see Supplementary Figure S7) and a water bath for heating reduction were installed in front of the quartz window of the reactor. The temperature of the reactor during the experiment was kept within 25–28 °C.

Phenol concentration was determined by high-performance liquid chromatography (RP, C18 Zorbax Eclipse, methanol/water 50/50, ~1 mL/min, MWD detector), and the obtained value was averaged over three probes.

### 3.3. Work Function Measurements

Thermal work function was determined by the Kelvin probe method (vibrating capacitor) with the SKP5050 instrument (KP Technology, Wick, United Kingdom) using a 2 mm gold tip and a gold reference to estimate the work function value of the tested sample. Solid-phase samples were pressed into tablets and sintered at a temperature of 1000 °C for 5 h. Sintered pellets were then used as one of the capacitor plates. The gradient value was set the same for all samples in the series. The experimental deviation of work function values at different points of the pellet did not exceed 15–20 mV.

### 3.4. Diffuse Reflectance Spectroscopy

Diffuse reflectance spectra were recorded at room temperature on a Cary 5000 spectrophotometer equipped with a 2500 DRA attachment for diffuse reflectance measurements. Optically pure $BaSO_4$ was used as a reference sample. Diffuse reflectance spectra of the pristine and doped samples are shown in Supplementary Figure S8. Application of the Tauc plot approach for diffuse reflectance spectra treatment allowed us to determine the apparent optical bandgap for all synthesized samples, which was estimated to be $3.05 \pm 0.07$ eV and corresponds well to the bandgap of rutile [42]. Therefore, $TiO_2$ doping with either Sc or Nb does not change the optical bandgap of $TiO_2$.

Experiments on photoinduced defect formation were carried out using sample irradiation with a 120 W high-pressure mercury lamp (DRK-120) and a UFS-2 bandpass filter (see Supplementary Figure S7). Irradiation intensity was 50 $\mu W/cm^2$ at 365 nm. Different diffuse reflectance spectra, $\Delta R$, were taken as a measure to estimate the absorption, $\Delta A$, of photoinduced defects:

$$\Delta A = \Delta R = R_0 - R_{h\nu}(t)$$

here, $R_0$ is a diffuse reflectance spectrum of the sample before irradiation and $R_{h\nu}(t)$ is a diffuse reflectance spectrum of the sample after UV irradiation for the time, $t$ (see Supplementary Figure S1).

### 3.5. EPR Measurements

Electron paramagnetic resonance spectra were recorded by Adani SPINSCAN X (Minsk, Belarus) (X-band). Cells filled with a certain amount of the sample were placed into the resonator without any preliminarily treatment. Spectra were measured in the air atmosphere. Typical instrument settings were as follows: frequency and modulation amplitude were 100 kHz and 20–75 $\mu T$, respectively, and the scanning rate was 0.4 Gauss/s. A Dewar flask was used for the measurements at a liquid nitrogen temperature of 77 K. Photoinduced

formation of paramagnetic defects was studied with sample irradiation by a mercury lamp (DRK-120) equipped with a UFS-2 bandpass filter (see Supplementary Figure S7), with simultaneous rotation of the cell by an overhead stirrer to achieve a uniform irradiation.

**Supplementary Materials:** The following supporting information can be downloaded at: https://www.mdpi.com/article/10.3390/catal12050484/s1, Description of photoinduced defect formation, Figure S1: Diffuse reflectance spectra before and after UV irradiation of undoped and doped $TiO_2$; Figure S2: EPR spectra of pristine and doped $TiO_2$; Figure S3: Integrated EPR spectra of Sc-doped $TiO_2$; Figure S4: XRD pattern of Sc- and Nb-doped $TiO_2$ samples; Figure S5: Electron microscopic images of undoped and doped $TiO_2$; Figure S6: XPS spectra of Sc- and Nb-doped $TiO_2$ samples; Figure S7: Transmittance spectra of BS-6 cut-off and UFS-2 bandpass filters; Figure S8: Diffuse reflectance spectra and Tauc plot transformation of Sc- and Nb-doped $TiO_2$ samples.

**Author Contributions:** Conceptualization, A.V.E. and D.W.B.; methodology, A.V.E.; software, P.D.M.; validation, A.V.R. and A.V.E.; formal analysis, P.D.M.; investigation, P.D.M.; resources, D.W.B.; writing—original draft preparation, P.D.M., A.V.R. and A.V.E.; writing—review and editing, A.V.E. and D.W.B.; visualization, P.D.M. and A.V.E.; supervision, D.W.B.; project administration, A.V.E.; funding acquisition, A.V.E. and D.W.B. All authors have read and agreed to the published version of the manuscript.

**Funding:** The reported study was funded by the Russian Foundation for Basic Research, project number 18-29-23035 mk, and supported by Saint Petersburg University (ID: 91696387).

**Data Availability Statement:** Data are contained within the article and supplementary material.

**Acknowledgments:** P.D.M., A.V.R. and A.V.E. are grateful to the Russian Foundation for Basic Research for financial support of the present study (Grant No: 18-29-23035 mk). All experiments were performed in the Laboratory "Photoactive Nanocomposite Materials" supported by Saint Petersburg University (ID: 91696387). The authors are grateful to the researchers of the resource centers "Nanophotonics", "Centre for X-ray Diffraction Studies", "Centre for Physical Methods of Surface Investigation", "Centre for Diagnostics of Functional Materials for Medicine, Pharmacology and Nanoelectronics", "Centre for Geo-Environmental Research and Modelling (GEOMODEL)", "Chemistry Educational Centre", and "Centre for Innovative Technologies of Composite Nanomaterials" of the SPbU Research Park for their excellent expertise and technical support in performing various sample characterizations.

**Conflicts of Interest:** The authors declare no conflict of interest.

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
