# Peer review of "Effect of the Heterovalent Doping of TiO2 with Sc3+ and Nb5+ on the Defect Distribution and Photocatalytic Activity"

_catalysts, doi:10.3390/catal12050484_

Round 1

Reviewer 1 Report

The article presents a very relevant topic and investigates the effects of the heterovalent doping with Sc3+ and Nb5+ in the TiO2 structure, focusing in the material photoactivity and defect distribution. The work’s purpose, background information and main contribution of the research are well presented in the Introduction section. The details of the methodology applied in the work (synthesis, photocatalytic tests, materials characterization, etc.) are satisfactorily described in the Materials and Methods section.

About the Results and Discussion section:

-I suggest comparing the photocatalytic results presently obtained with previous works described in the literature. Other authors have already studied TiO2-based photocatalysts doped with Sc or Nb. Did they observed similar trends?

-There are cases in which the anatase phase is more photocatalytically active than rutile. Why did the authors choose to study only the rutile phase in the present work? Presenting what motivated this choice can be quite enriching for the work.

-Some works have already described that TiO2 doping may increase in the visible light absorption (10.1134/S0020168513040134, 10.1016/j.ceramint.2015.12.029, 10.1016/j.apsusc.2015.12.166). What have the authors find in this regard?

-Cui et al. (1995) have pointed the optimal concentration of Nb2O5 on TiO2 in the range between 1.5% and 3.0% (doi: 10.1006/jssc.1995.1118). Joskowska et al. (2010) studied the photoconductivity of Nb2O5/TiO2 sol–gel thin films, and observed a higher photoconductivity in the sample with 20% Nb (10.1016/j.jnoncrysol.2010.05.075). Authors should mention and discuss it.

-Regarding the XRD results, authors should identify the phase and reflection planes in the diffractograms, and indicate the card number of the identified phase, according to a XRD database.

-In the Materials and Methods section, authors mention the Rietveld refinement methodology. Authors should present the obtained results; Da Silva et al. (2017, 10.1016/j.apsusc.2016.09.126), observed the increase of the cell parameters after TiO2 was doped with Nb, which was considered as an evidence of the incorporation of Nb5+ in the TiO2 lattice. A similar trend was observed by the authors?

Based on the article analysis, and considering the relevance of the research for the field of catalysis, I recommend the article for publication after minor revisions.  

Author Response

We are very grateful to reviewers for the attentive and positive consideration of our manuscript and for the useful comments and suggestions that help us to enhance the scientific merits of the manuscript.

Reviewer 1

The article presents a very relevant topic and investigates the effects of the heterovalent doping with Sc3+ and Nb5+ in the TiO2 structure, focusing in the material photoactivity and defect distribution. The work’s purpose, background information and main contribution of the research are well presented in the Introduction section. The details of the methodology applied in the work (synthesis, photocatalytic tests, materials characterization, etc.) are satisfactorily described in the Materials and Methods section.

We are grateful for the positive consideration of our manuscript.

-I suggest comparing the photocatalytic results presently obtained with previous works described in the literature. Other authors have already studied TiO2-based photocatalysts doped with Sc or Nb. Did they observe similar trends?

A brief description of the results related to the effect of doping with either Sc3+ or Nb5+ reported by other research groups is given in Introduction section (refs. 15 – 30). The major problem of the corresponding studies is that they were performed either without a systematic approach or for a limited range of the dopant contents. However, in general, one may conclude that a tendency of volcano like dependence of photocatalytic activity on the dopant concentration is rather a typical one. It is also wise to note that most studies of the dopant effect on photocatalytic activity were done for anatase phase of titania.

-There are cases in which the anatase phase is more photocatalytically active than rutile. Why did the authors choose to study only the rutile phase in the present work? Presenting what motivated this choice can be quite enriching for the work.

We are well aware in the existence of the general point of view that anatase is more active than rutile. We also familiar with the hypothesis that it can be explained considering the difference in band gap and conduction band position between anatase and rutile. At the same time, this generalization contradicts to the experimental results quite often when some particular rutile samples demonstrate higher activity then anatase samples. In fact, the photocatalytic activity depends stronger on synthesis method and particle morphology rather than on phase composition of the titania samples. Currently we perform similar systematic studies of the dopant effect on photoactivity of anatase TiO2. Generally, it is known that the effect of the same type of dopant on defect formation and its photocatalytic behavior is rather quite different for anatase and rutile. Thus, it is important to understand the effect of the heterovalent doping for both phases. Here we report our results obtained for rutile phase.

-Some works have already described that TiO2 doping may increase in the visible light absorption (10.1134/S0020168513040134, 10.1016/j.ceramint.2015.12.029, 10.1016/j.apsusc.2015.12.166). What have the authors find in this regard?

We describe the results obtained by diffuse reflectance spectroscopy to estimate the values of the optical band gaps for pristine and doped samples in the section Materials and Methods. As evident from the results presented in Figure S8 TiO2doping with either Sc or Nb does not change the optical band gap of TiO2 and estimated optical band gap for all samples is 3.05 ± 0.07 eV that corresponds well to the optical band gap of rutile.

-Cui et al. (1995) have pointed the optimal concentration of Nb2O5 on TiO2 in the range between 1.5% and 3.0% (doi: 10.1006/jssc.1995.1118). Joskowska et al. (2010) studied the photoconductivity of Nb2O5/TiO2 sol–gel thin films, and observed a higher photoconductivity in the sample with 20% Nb (10.1016/j.jnoncrysol.2010.05.075). Authors should mention and discuss it.

Here we report the results of the systematic studies of the dopant content within the concentration range 0 – 1,7 at.% on photocatalytic activity and its relation with the defect redistribution caused by doping. In general, doping change many parameters of the original materials since basically it affects its chemical composition and therefore, its physical properties.

The work by Cui et al. reports the effects of Nb doping on photocatalytic activity and surface acidity and there is no apparent correlation between the alteration of these two properties of doped TiO2. Indeed, the photocatalytic activity demonstrates a volcano like dependence on the dopant content whereas the surface acidity trend demonstrates Z-like behavior. Besides, there is no report about phase composition of the samples. Considering that the original sample was P25 and the calcination temperature was 550 C one can assume that the major phase the samples were of the complex phase compositions containing both anatase and rutile without a known proportion between phases, which can be different due to effect of different dopant content. Thus, these results cannot be considered as truly reliable at present state of scientific research.

The work by Joskowska et al. reports the effect of heavy Nb doping on photoelectrical behavior of TiO2. First, the degree of doping is very high and basically one can consider the samples rather as solid solutions than as doped samples. Second, in general, there is not direct connection between electrophysical and photocatalytic behavior of photocatalysts. Therefore, these data are not relevant to the results reported in our manuscript.

Therefore, we do not feel that the results reported in these two papers suggested by reviewer are significant for the present studies and should be mentioned and discussed in our manuscript.

The work by  

-Regarding the XRD results, authors should identify the phase and reflection planes in the diffractograms, and indicate the card number of the identified phase, according to a XRD database.

The XRD spectra of rutile are well known and have been reported uncountable times in the literature. Thus, we do not feel it would be useful to analyze and report them in details once again. As mentioned in the Materials and Methods section the phase composition was determined using the TOPAS 4.2 program (Bruker AXS) linked to ICSD database.

-In the Materials and Methods section, authors mention the Rietveld refinement methodology. Authors should present the obtained results; Da Silva et al. (2017, 10.1016/j.apsusc.2016.09.126), observed the increase of the cell parameters after TiOwas doped with Nb, which was considered as evidence of the incorporation of Nb5+ in the TiO2 lattice. A similar trend was observed by the authors?

We have observed an increase of the cell volume with the increase of the dopant content for Nb doped samples (see Figure below). Since the observed effect can be considered as rather macroscopic and it does not correlate with the dependence of photocatalytic activity on the dopant content, we treat this observation as non-relevant to the discussed issues.

Based on the article analysis, and considering the relevance of the research for the field of catalysis, I recommend the article for publication after minor revisions.  

Thank you once again for your attentive and positive consideration of our manuscript.

Reviewer 2 Report

This is a well-written manuscript with solid evidence showing that the photocatalytic activity is dependent on the work function and the defect states of the doped-TiO2 materials, which provides insights in understanding the effect of dopants in TiO2 on its defect states and activity. The methods are described in detail and the experiments well support authors' claim. Thus I recommend the publication after the following issue being addressed: "Fig. 6B and its description are not consistent. Please revise the corresponding description and the discussion part. It is unclear what the difference refers to in Fig. 6B."

Author Response

We are very grateful to reviewers for the attentive and positive consideration of our manuscript and for the useful comments and suggestions that help us to enhance the scientific merits of the manuscript.

Reviewer 2

This is a well-written manuscript with solid evidence showing that the photocatalytic activity is dependent on the work function and the defect states of the doped-TiO2 materials, which provides insights in understanding the effect of dopants in TiO2 on its defect states and activity. The methods are described in detail and the experiments well support authors' claim. Thus, I recommend the publication after the following issue being addressed: "Fig. 6B and its description are not consistent. Please revise the corresponding description and the discussion part. It is unclear what the difference refers to in Fig. 6B."

We are thankful to Reviewer for the positive consideration of our manuscript and for indicating of the problematic issue with the Figure 6. The Y-axis inscription in Figure 6b has been corrected. A description and discussion of the EPR data for Nb-doped TiO2 is given immediately below Figure 6 (p. 7), and it is also used in the discussion of defect redistribution as a function of Nb dopant concentration on page 8. We have verified that currently there is no discrepancy between the description in the text and the data in Figure 6.

Reviewer 3 Report

The manuscript reported interesting results about the Sc/Nb doped rutile TiO2 photocatalysts. Also the promotion of photocatalytic activity was reasonably explained. It could be accepted for publication after minor revision.

Some comments:

  1. The Figures could be reorganized to be more compact.
  2. Same fonts were recommended in the same Figure.
  3. Does there exist the charge transfer between phenol and doped TiO2? and it affects the photocatalytic activity?
  4. Why two curves in Figure S1b?

Author Response

We are very grateful to reviewers for the attentive and positive consideration of our manuscript and for the useful comments and suggestions that help us to enhance the scientific merits of the manuscript.

Reviewer 3

The manuscript reported interesting results about the Sc/Nb doped rutile TiO2 photocatalysts. Also the promotion of photocatalytic activity was reasonably explained. It could be accepted for publication after minor revision.

Some comments:

  1. The Figures could be reorganized to be more compact.

Unfortunately, we do not see how the figures could be reorganized without loosing their clearness and informativeness.

  1. Same fonts were recommended in the same Figure.

Thank you! Fonts have been corrected.

  1. Does there exist the charge transfer between phenol and doped TiO2? and it affects the photocatalytic activity?

To avoid direct photoexcitation of phenol and therefore, charge transfer process, we used cut-off filter BS-6 with ~350 nm transmission limit. Therefore, we assume that there is no effect of the charge transfer process on photocatalytic activity of doped TiO2.

  1. Why two curves in Figure S1b?

Thank you! An assignment of the curves in Figure S1b to pristine and niobium-doped TiO2 samples has been done.

Reviewer 4 Report

In this manuscript, the authors described a study of doping effect with Nb5+ and Sc3+ in TiO2 photocatalysts. The authors described the optimal dopant concentrations and a redistribution of the defect states in TiO2. Overall, the manuscript is well written however, I do have one major concern regarding the discussion on Volcano like plot in the introduction.

In photocatalysis, doping could shift the fermi level and thus making semiconducting materials n-type of p-type. When in contact with electrolyte, the n,p-dopants could induce interfacial electric field. Such a field could then be used to separate charge carriers to promote photocatalysis. And if the light penetration depth matches the field depth, photocatalysis could reach maximum efficiency. The depth of the field depends highly on the dopants concentration, which could also give a Volcano like plot presented in the manuscript. The author seems not considering this effect in the current discussion and I suggest the authors to add a discussion with dopant concentration and field strength as one possible cause for the volcano plot with relevant references.

Author Response

We are very grateful to reviewers for the attentive and positive consideration of our manuscript and for the useful comments and suggestions that help us to enhance the scientific merits of the manuscript.

Reviewer 4

In this manuscript, the authors described a study of doping effect with Nb5+ and Sc3+ in TiO2 photocatalysts. The authors described the optimal dopant concentrations and a redistribution of the defect states in TiO2. Overall, the manuscript is well written however, I do have one major concern regarding the discussion on Volcano like plot in the introduction.

Thank you very much for your positive consideration of our manuscript!

In photocatalysis, doping could shift the fermi level and thus making semiconducting materials n-type of p-type. When in contact with electrolyte, the n,p-dopants could induce interfacial electric field. Such a field could then be used to separate charge carriers to promote photocatalysis. And if the light penetration depth matches the field depth, photocatalysis could reach maximum efficiency. The depth of the field depends highly on the dopants concentration, which could also give a Volcano like plot presented in the manuscript. The author seems not considering this effect in the current discussion and I suggest the authors to add a discussion with dopant concentration and field strength as one possible cause for the volcano plot with relevant references.

In our study we observed the qualitative correlations between volcano like dependencies of photocatalytic activity, work function and intrinsic defect concentrations. It is wise to note that work function was measured by Kelvin probe method at ambient atmosphere without the presence of electrolyte (phenol solution). The same is true for the spectroscopic data for alteration of the intrinsic defect concentrations as a function of dopant content. Thus, we can assume that a different environment, either atmospheric or liquid solution, is not a factor responsible for the observed correlations. In fact, both correlations between defect distribution and work function, and between defect distribution and photocatalytic activity, indicate that the major reason for the alteration of both the work function and photocatalytic activity and for the correlation between these two parameters is an intrinsic defect redistribution caused by dopants. Consequently, a possible formation of the subsurface electric field and its correlation with a light penetration depth is not a (major) factor, which determines a volcano like character of the dependence of photocatalytic activity on the dopant concentration.

We have done our best to improve the English and the overall quality of our manuscript.
